# Repair Foci as Liquid Phase Separation: Evidence and Limitations

**DOI:** 10.3390/genes13101846

**Published:** 2022-10-13

**Authors:** Judith Miné-Hattab, Siyu Liu, Angela Taddei

**Affiliations:** Institut Curie, PSL University, Sorbonne Université, CNRS, Nuclear Dynamics, CEDEX 05, 75248 Paris, France

**Keywords:** DNA repair, liquid phase separation, double strand break

## Abstract

In response to DNA double strand breaks (DSB), repair proteins accumulate at damaged sites, forming membrane-less condensates or “foci”. The formation of these foci and their disassembly within the proper time window are essential for genome integrity. However, how these membrane-less sub-compartments are formed, maintained and disassembled remains unclear. Recently, several studies across different model organisms proposed that DNA repair foci form via liquid phase separation. In this review, we discuss the current research investigating the physical nature of repair foci. First, we present the different models of condensates proposed in the literature, highlighting the criteria to differentiate them. Second, we discuss evidence of liquid phase separation at DNA repair sites and the limitations of this model to fully describe structures formed in response to DNA damage. Finally, we discuss the origin and possible function of liquid phase separation for DNA repair processes.

## 1. Introduction on DNA Repair of Double Strand Break

Throughout the life of a cell, its genome is exposed to a large variety of DNA damages, including DNA base lesions, inter-strand crosslinks, single or double-strand breaks (DSBs). It is estimated that the DNA of a human cell is subject to 50,000 to 500,000 lesions per day [1]. Among the different kinds of DNA damages, double strand breaks (DSB) represent only a very small fraction of DNA lesions: it is estimated that a human cell experiences around ~50 DSBs each day [2]. Despite their low number, DSBs are extremely toxic to the cells as they fragment chromosomes into pieces (which if not connected with the centromere will be mis-segregated upon cell division). DSBs can be provoked by endogenous natural stress during replication or programed events during meiosis, immune system development or by exogenous stress such as radiation, chemotherapeutics or crosslinking agents. Healthy cells are able to repair DNA damages and preserve their genomic integrity. However, the accumulation of DNA damages or improper repair can lead to mutations, loss of genetic information and cancer [3]. In higher eukaryotes, germline mutations in DNA repair genes lead to diseases such as Werner, Bloom syndromes and cancer. Thus, DNA repair is an essential process for preserving genome integrity [3]. Recent studies have revealed that some repair proteins accumulate at DNA damaged sites by forming a liquid–liquid phase separation (LLPS). In this review, we describe the current understanding of LLPS in the context of DNA repair. After introducing the different mechanisms of DSB repair, we will present several models of condensates including the LLPS model, and we will highlight the criteria to differentiate them. We will then discuss evidence of LLPS in the context of DNA repair and the limitation of this model to fully describe structures formed at damaged sites. Finally, we will discuss the origin and the proposed functions of LLPS.

### 1.1. Mechanisms of DSB Repair 

Eukaryotic organisms use three mechanisms to repair DSBs: canonical non-homologous end-joining (c-NHEJ), homologous recombination (HR) and alternative non-homologous end joining (alt-NHEJ). The main steps and proteins of these pathways are conserved from bacteria to humans (Figure 1). However, their relative use varies between species: HR predominates in *S*. *Cerevisiae (Sc.)* yeast, whereas c-NHEJ contributes significantly to DSB repair in vertebrates. When c-NHEJ or HR are defective, Alt-NHEJ acts as a back-up pathway [4]. C-NHEJ and HR have different kinetics: c-NHEJ is a fast process that can be completed in approximately 30 min in human cells, whereas HR can take 7 h or longer to complete [5]. In the following sections, we summarize the main steps of each pathway.

Following the induction of DNA damage, the immediate reaction of a cell is to detect various DNA lesions using DNA damage sensors. These sensors are abundant in the nucleus. In response to DNA damage, they are essential to initiate signals to recruit DNA damage repair (DDR) factors and activate other relevant biological processes such as cell cycle arrest. In human cells, one of the earliest post-translational modifications appearing at the DNA lesions is poly-ADP-ribose (PAR), a NAD+ dependent modification carried out by PAR polymerases (PARPs) [6]. PARP1, the founding member of the PARP enzyme family, is recruited to sites of DNA damage as early as 1 s after DNA damage and locally produces the ADP-ribose polymer [7]. PARP1 is one of the most abundant nuclear polypeptides, with an estimated 1–2 million molecules per nucleus [8]: these molecules scan nucleosomes of chromatin to quickly detect DNA damage. Each damaged DNA site is bound by 1 to 2 PARP1 molecules [9]. Most PAR chains are attached to PARP1 itself, but a number of other proteins are also targets of PARylation, among them histones [10]. Early PARylation plays a central role in DNA repair: Indeed, removing its pioneering factor PARP1 or inhibiting its catalytic activity strongly impact the recruitment of many repair proteins, which ultimately leads to a dramatic decrease in the DNA repair efficiency [11]. Chromatin decompaction is detected at sites of DNA damages in a PARP-dependent manner, making the damaged area more accessible for the recruitment of a cascade of repair proteins [12]. Among them, FUS, EWS and TAF15, belonging to the FET-protein family of RNA and DNA binding proteins, are recruited in a PARP-dependent manner at sites of DSB repair [12,13]. These proteins have an emerging role in DSB repair [14], but their precise action is not fully understood. EWS fusion is at the origin of pediatric tumors [15]; its depletion reduces cell viability upon UV-irradiations [16] and Ewing sarcoma cells are sensitive to PARP inhibitors [17]. The FUS (FUsed in Sarcoma) protein also plays a role emerging at different steps of DSB repair, both by c-NHEJ and HR [14]. FUS proteins bind PAR chains generated by PARP1 [14]. It was shown that FUS depletion leads to a decrease in H2AX phosphorylation and an impairment in DSB repair foci formation such as 53BP1 and Ku70 [14]. Around 50 s after DNA damage, PARG (poly ADP-ribose glycohydrolase), the antagonist protein of PARP1, starts dePARylation, i.e., the removal of PAR chains. The FUS proteins are then replaced by 53BP1 (p53 binding protein 1), a protein paying an essential role in orchestrating the choice of the DSB repair pathway. Both PARP1 and FUS also have a role during later stages of DSB repair by the different pathways [14]. Then, Ku70/80, the ATM kinases that phosphorylates histone variant H2AX to generate γH2AX is recruited, as well as the MRN complex (MRE11, RAD50 and NBS1), which bind directly to the DNA ends and arrive ∼13 s after DSB [6]. While RAD50 recognizes naked DNA ends and holds them in close proximity to each other, MRE11 is an endo- and exo-nuclease that processes DNA ends prior to religation for resection-dependent c-NHEJ [18] or prior to more extensive resection by later nucleases for HR. Resection at DSBs by either Ku70/80 or MRN plays a central role in determining the outcome of the competition between the different repair pathways [19]. In *Sc.* yeast, PARylation does not occur and the first repair proteins detectable at the sites of DSB are the yKu (yKu70/yKu80) and MRX (Mre11-Rad50-Xrs2) complexes, which bind directly to DNA ends [20].

In human cells, c-NHEJ appears to repair nearly all DSBs outside of S-G2 cell cycle phases and even approximately 80% of DSBs within S-G2 that are not proximal to a replication fork [21]. Thus, c-NHEJ represents the major pathway for the repair of DSBs. When the c-NHEJ pathway is chosen to repair DSB, the two DNA ends are religated after minimal DNA end-processing to trim or fill in the ends to make them optimally ligatable by the DNA ligase IV complex [21]. The general mechanism of c-NHEJ can be divided into several steps, which are: (i) DNA end recognition and stabilization of the c-NHEJ complex by the Ku complex; (ii) DNA end processing by several c-NHEJ proteins, including Artemis, PNKP, APLF, Polymerases μ and λ, Werner (WRN), aprataxin and Ku; and (iii) ligation of the broken ends by DNA ligase IV and XRCC4, which stimulates its ligation activity [22].

HR is a more complex pathway: it occurs primarily in S-G2 phase cells and uses an undamaged homologous DNA sequence as a template for copying the missing information. HR is a highly conserved process during evolution and promoted by enzymes of the Rad52 epistasis group, including Rad51, Rad52, Rad54. When a DSB forms and HR is chosen to repair the break, HR occurs in several steps, including resection (more than 100 bases), formation of a nucleo-filament, homology search/strand invasion and D-loop resolution. First, the 5′ ends of the DNA break are resected by nucleases to yield 3′ single-stranded DNA (ssDNA) tails [23]. This ssDNA is bound by replication protein A (RPA) to control its accessibility to the next repair proteins of the Rad52 epistasis group. The Rad52 proteins then stimulate the removal of RPA and recruit the recombinase Rad51 to DNA, which polymerizes on the broken DNA ends [24]. The Rad51 paralogs RAD51BRAD51C-XRCC2-XRCC3 (Rad55-Rad57 in yeast) and RAD51-DXRCC2-SWS1 (the Shu complex Psy3- Csm2-Shu1-Shu2) stabilize Rad51 filaments on ssDNA. The Rad51-DNA complex, called the nucleo-filament, has the capacity to search throughout the genome, identify a region of homology and then to promote strand invasion of the homologous duplex DNA. At this stage, anti-recombination functions are exerted by FBH1, PARI (Srs2), which dissociates Rad51 nucleo-filaments from ssDNA, and FANCM (Mph1), which disassembles D-loops [19]. During HR, a common source for the duplex DNA donor is the undamaged sister chromatid; however, homologous sequences on either the homolog or on a different chromosome can be captured by the presynaptic nucleofilament to perform inter-homolog recombination or ectopic recombination, respectively. While genomic regions overlapping in the nuclear space recombine more efficiently than sequences located in spatially distant territories [25], pairing between distant loci can be detected as early as 2 h after damage in *Sc.* yeast [26]. Ectopic recombination is thought to be rare in the mammalian genome, possibly owing from the efficiency of NHEJ and the large size of the genome. However, large-scale analysis of human samples recently unveiled that inter-chromosome somatic recombination of repeat elements is widespread and tissue specific [27]. Exploring the megabases (Mbs) of the genome and the nuclear space for homologous sequence is a limiting step in the case of ectopic recombination that remains mysterious [28]. Once homology is found, the invading strand primes DNA synthesis of the homologous template, ultimately restoring genetic information disrupted at the DSB.

When NHEJ or HR are defective, Alt-NHEJ acts as a back-up pathway [4]. In addition, it was proposed that Alt-NHEJ is activated upon certain forms of DSBs with damaged termini, such as those induced by IR [29], and plays a pivotal role in DSB repair of mitochondrial DNA [30]. Alt-NHEJ repair comprises two sub-pathways that differentially operate on the basis of the DNA sequence complementarity degree at the DNA ends of DSBs: microhomology-mediated end joining (MMEJ), which requires from 2 to 20 nucleotides of sequence homology and a less characterized end joining (EJ) pathway involving very little sequence homology at the repair site [31]. MMEJ is the better characterized sub-pathway of Alt-NHEJ, and most of the reviews illustrate only MMEJ to refer to as Alt-NHEJ. The first step of MMEJ requires DNA end resection by the MRN/CtlP complex, specifically recruited to DSB by poly (ADP ribose) polymerase-1 (PARP-1). CtIP activation enhances the MRN endonuclease/exonuclease activity [32] resulting in exposition of the microhomology sequence within single strand regions. The next step is the bridging and alignment of the DNA ends via the short microhomologies, operated in conjunction by PARP-1, MRN and PolQ activity. Then, Non-homologous 3′ tails are digested by ERCC1\XPF nucleases, generating gaps within DNA strands, which are filled by PolQ-mediated DNA synthesis. In the last step, DNA DSBs are finally repaired by DNA ligase 3 (LIG3)/XRCC1 complex [31].

### 1.2. Repair Condensates Are Formed in Response to DNA Repair 

In response to DNA damage in eukaryotes, many repair proteins relocalize from a diffuse nuclear distribution to nuclear foci of high local concentration at the site of the DNA lesion [33]. Those protein centers can be visualized as fluorescent foci of tagged repair proteins in living cells [34,35,36] or by immunofluorescence [34,37]. The current number of molecules required for processing a single DNA break is not always known and depends on the repair protein; however, it is clear that even a single DSB can induce the formation of a visible focus by fluorescent microscopy [34,36]. For example, in haploid yeast, a single DSB is sufficient for the formation of a visible Rad52 focus containing 600–2100 molecules out of approximately 2300 molecules [38]. Such repair focus exhibits a higher local concentration of Rad52 at the DSB relative to its diffuse nuclear distribution in undamaged cells. Of note, some repair proteins do not accumulate as a focus visible by fluorescent microscopy at DSB in vivo. Some proteins interact too transiently to manifest as foci (such as the signaling kinases CHK1 and -2 in human cells [34]) or assemble at DSB loci in a focus containing very few molecules hardly detectable from the background (DNA-PK/Ku70 in mammalian cells [34], the SHU complex in yeast [33]). Of note, foci observed experimentally are often formed at artificial DNA breaks. When induced with a restriction enzyme, both sisters are likely cut, making them difficult to repair and allowing more time for proteins to accumulate into foci. When induced by irradiation or by mutagenic drugs, the repair foci observed may result from the clustering of several DNA breaks. However, yeast HR proteins, including Rad52, Mre11 or Sae2, form foci of similar appearance whether they result from a spontaneous event or a restriction enzyme induced break [36].

Repair foci can assemble and disassemble within minutes [36,39]: Such dynamics are essential to allow the turnover of the different repair proteins involved in DSB repair. Additional dynamics are likely provided by post-translational modifications (PTMs), which are continuously applied and erased during repair [33]. Recent studies also investigated the internal dynamics of repair foci. Some HR proteins inside foci rapidly exchange with the nuclear proteins, such as Rad52 in yeast [40], RAD52 and RAD54 in mammalian cells [35], while other HR proteins appear stably associated with repair foci (RAD51 in mammalian cells [35] or Rfa1 in yeast [40]). With the arrival of single molecule microscopy, it is now possible to precisely address the internal structure and dynamics of repair condensates, opening new avenues to shed light on their physical nature.

## 2. Models of Condensates

The mechanisms driving the formation, maintenance and disassembly of repair condensates is not fully understood. An emerging hypothesis in the field of cell biology is that some repair condensates are formed via liquid–liquid phase transition (LLPS) [41]. However, the physical nature of repair condensates can vary for each protein: in the following section, we present three models of condensates, as well as criteria and experimental methods to differentiate them.

### 2.1. Different Models of Membrane-Less Sub-Compartments

First, in a simple model named “**binding model**”, condensates can arise through the direct binding of proteins to specific sites on a nucleic acid. Such condensates do not exhibit any phase separation and can lead to a high concentration of protein only if the binding sites themselves are concentrated. Their existence relies only on the protein’s binding properties and on the number of binding sites on chromatin. However, this type of subnuclear compartment could sequester specific factors because of the high avidity resulting from the density of binding sites.

Second, if in addition, proteins can bridge distant chromatin loci by creating loops, a polymer–polymer-phase separation (**PPPS**) can emerge. To create a polymer phase separation, the density of “bridges” must be sufficiently high so the nucleic acid transitions from an extended coil to a collapsed globule [42]. These interactions can be driven by specific or multivalent weak interactions between chromatin binding proteins or chromatin components [43]. In this model, the existence of condensate relies on both the binding and bridging properties of the proteins forming the PPPS.

Third, an alternative attractive hypothesis is that membrane-less condensates arise from a liquid–liquid phase separation (**LLPS**) forming droplets [41]. LLPS is a thermodynamic process by which, above a threshold macromolecule concentration, two coexisting liquid phases form to minimize free energy. Unlike the first two models, in the droplet model, proteins self-organize into liquid-like droplets that grow around the chromatin fiber, allowing certain molecules to become concentrated while excluding others. Importantly, unlike the binding and the PPPS models, in the LLPS model, the large majority of molecules inside the droplet are not physically bound to a polymer substrate.

Thus, an important difference between the LLPS and PPPS/binding model is that once formed, condensates formed via LLPS can persist independently of chromatin, while condensates formed via binding/PPPS strictly rely on chromatin and would disassemble in its absence. LLPS is generally attributed to the synergy of weak multivalent interactions among intrinsically disordered regions (IDRs) in proteins and nucleic acids. Within an LLPS, certain proteins and nucleic acids function as **scaffolds** and are necessary for organizing the formation of specific condensates [44]. Scaffold proteins recruit **client** molecules, which are not sufficient for condensate formation on their own but associate with condensates due to direct interaction with scaffolds. 

### 2.2. Criteria to Define PPPS and LLPS

In the recent years, LLPS has received so much attention that it has almost become the default explanation for all membrane-less condensates without objective criteria. In addition, several experimental artefacts sometimes led to hasty conclusions on the physical nature of condensates [45]. Importantly, there is often a confusion between PPPS and LLPS: one should keep in mind that a phase separation can emerge from polymers without necessarily being in a liquid phase. In that case, proteins belonging to a PPPS have a very similar macroscopic appearance by fluorescent microscopy and can easily be mistaken as an LLPS. In the following section, we summarize several macroscopic and microscopic criteria used in the literature to properly define an LLPS (see Figure 2).

#### 2.2.1. “Standard Criteria” in the Literature to Demonstrate That a Condensate Forms via LLPS

#####  **Maintenance of a Spherical Shape**

The first criteria used to describe an LLPS is its spherical aspect [41]. The shape of a liquid phase is typically dominated by surface tension, which leads to a spherical shape. Surface tension is a mechanical tension that exists at the boundary between two phases. It tends to reduce the area of the interface until it reaches a minimum. The minimum area of a drop corresponds to a spherical shape; therefore, surface tension drives liquid drops to be spherical. However, the quantification of the condensate’s roundness can be challenging in tiny bacteria or in yeast using conventional microscopy. For example, using super resolution microscopy (live photo activable localization microscopy), it was shown that the radius of Rad52 foci in *Sc*. yeast is 120 nm, a factor of 2 below the diffraction limit [40]. Such small foci would always appear spherical when imaged through conventional microscopy.

Although the spherical shape is commonly used as a strong criterium to assess the liquid nature of condensates, it is not a sufficient condition to define an LLPS. Indeed, alternative models such as PPPS can also give rise to condensates with spherical shapes. For example, single molecule microscopy reveals that despite their spherical shape, replication compartments and paraspeckles are not consistent within an LLPS [45,46,47,48].

#####  **Wetting Behavior**


In addition to their spherical shape, LLPS are expected to exhibit wetting behavior, i.e., flattening against an obstacle (e.g., the nuclear/cell envelop). While the sphericity of condensates is commonly quantified, the wetting behavior is rarely tested, maybe because these events are less frequent and rare to observe within a timelapse movie [49].

#####  **Fuse after Touching**


An important hallmark of LLPS is their ability to fuse, two droplets for radius R leading to a bigger droplet of doubled volume. The fusion of repair foci was previously observed in yeast [49,50] and FUS, EWS, TAF15, 53BP1 and p53 in human cells [51,52,53,54] by following these foci in living cells at the minute time-scale. To test the fusion of condensates, it is important to differentiate a “clustering of solid condensates”, in which multiple condensates cluster but are close without merging, from a “real fusion” of droplets where molecules are shared between the different condensates diffusing within a larger LLPS. Furthermore, PPPS also shares the property of fusion if they are close enough. The observation of fusion events by conventional microscopy is often not sufficient to distinguish a clustering from a real fusion [45].

#####  **Mobility of the Molecules within the Condensate**


Watching how proteins move and interact within a condensate is crucial for better understanding their biological mechanisms. SPT is a powerful technique that makes these observations possible by taking “live” recordings of individual molecules in a cell at high temporal and spatial resolution (50 Hz, 30 nm) [55,56]. Based on the way individual molecules move in vivo, SPT allows for (i) sorting proteins into subpopulations characterized by their apparent diffusion coefficients, (ii) quantifying their motion, (iii) estimating residence times in specific regions of the nucleus, (iv) and testing the existence of a potential attracting or repelling molecules within distances smaller than the diffraction limit. In the case of condensate, SPT is particularly useful to measure the mobility of molecules within condensate or at their boundaries, as we will discuss below.

Inside an LLPS, most molecules are not physically bound to a polymer substrate. Rapid mobility is, thus, often proposed to discriminate LLPS from simple binding, and molecules within the droplet generally exhibit a higher diffusion coefficient than the few molecules bound to the substrate. Such behavior can be measured by FRAP (fluorescence recover after photobleaching) for large LLPS or by SPT (single particle tracking) for smaller LLPS. 

This criterium is necessary but not sufficient to confirm the liquid nature of condensates. For example, viral replication condensates (RCs) share several properties of LLPS such as high internal mobility and fusion: however, SPT experiments reveal that RNA PolII freely diffuses across these condensate boundaries, arguing that RCs do not consist of a distinct liquid phase [45]. In fact, higher mobility than the substrate could also be observed in certain cases in the PPPS model. For example, in the binding model, protein–DNA interactions can be extremely transient, so proteins stay most of the time unbound. If in addition, chromatin is very dense within the condensate, proteins can be “caged”. In that case, proteins will exhibit high internal dynamics such as the one predicted in LLPS.

#### 2.2.2. Evolving Metrics for LLPS

With the large number of studies claiming that condensates are LLPS with a lack of compelling evidence, the field is adopting more stringent criteria for classifying condensates as LLPS. The criteria below are used less often, because they are experimentally more difficult to access but are essential to distinguish LLPS from alternative models.

#####  **Concentration Dependence**


Each model predicts a distinct relationship between compartment size and component concentration. LLPS can buffer the nucleoplasmic concentration, while binding and bridging mechanisms cannot. Indeed, in the binding model, increasing the number of proteins will first increase the intensity of the focus. In the PPPS model, this will in addition reduce the size of the condensate as more bridges are formed, provoking chromatin to collapse more. However, after reaching a saturation point, the size and intensity of condensates will plateau in both cases and only the nucleoplasmic concentration outside will increase. Conversely, in the LLPS model, increasing protein amounts should increase the size of the droplet, and the concentration will remain constant both in the nucleoplasm and in the droplet. When an LLPS is formed but the observed molecule is not the main species driving the LLPS (referred to as a “client” of an LLPS), the concentration inside foci and in the background increases linearly with over-expression of the observed molecule [41].

#####  **Internal Mixing**


As discussed above, one feature of LLPS is the presence of a semi-permeable interface or boundary that is energetically unfavorable to cross. Monitoring internal mixing of molecules within the condensate by FRAP can be used to assess the permeability of condensate boundaries [57]. Although very effective, this criterium is restricted to condensates sufficiently large to be half bleached and nuclei with a large reservoir of proteins outside the condensate. In the extreme case of an impermeable boundary, fluorescence recovery of the bleached versus non-bleached half should be anti-correlated as the bleached area would recover only using molecules moving from the non-bleached half. On the contrary, in the absence of any boundary that would keep molecules within the condensate, the intensity in the bleached half would recover, whereas the intensity in the non-bleached half would only exhibit a subtle and transient intensity drop. The LLPS would correspond to an intermediate case of a semipermeable boundary, anti-correlated behavior of both halves would be observed while molecules mix internally, until transport across the boundary would lead to recovery of both halves. 

#####  **Diffusion across Boundary**


The arrival of single molecule microscopy opens new avenues to test the internal structure and dynamics of condensates at high resolution. In particular, SPT allows the tracking of individual molecules at a high resolution, and statistical analysis of their trajectories reveal their mode of diffusion. To move across the boundary of an LLPS, LLPS components must break cohesive interactions. As this is energetically unfavorable, molecules within an LLPS tend to bounce at the interface or move slowly across it. In contrast, there is no interface to impede diffusion inside or outside of a condensate formed by binding or bridging. Molecules move through these condensates with no change in the diffusion rate. Thus, measuring the change in diffusion coefficient of molecules traversing the foci boundary turns out to be a good criterion to distinguish binding/PPPS from LLPS [45]. 

The diffusive behavior of molecules at the condensate boundary can be measured by SPT. From the trajectories of individual molecules, it is possible to quantify the radial movement of all molecules as a function of their initial position relative to the center of the focus [40,58]. Concentrating on molecules close to the boundary, in LLPS, we expect to see a region of attraction with an average movement of molecules towards the center of the focus. Such behavior does not mean that this potential attracts molecules to the center of the focus, but it hinders the escape from the focus.

#####  **Type of Motion within the Condensate**


The type of motion that protein undergo within the condensate is indicative of its physical nature. In the binding/PPPS models, bound proteins can show two types of behavior: the bound fraction moving similar to the binding sites (anomalous sub-diffusion for chromatin) and the unbound fraction undergoing free diffusion [40]. This is all true in the limit where binding events happen on time scales much larger than the time-interval of measurements and when the binding sites are much less diffusive than the particles themselves. In contrast, within an LLPS, proteins should exhibit confined diffusion with a radius of constraint corresponding to the size of the droplet. Therefore, trajectories within an LLPS are more homogenous, a single mode of diffusion being usually sufficient to describe them. If trajectories are long enough, they should appear uniform with time in an LLPS, as long as the molecule does not reach the boundary of the condensate.

When multiple condensates fuse as expected for LLPS, a larger droplet is formed inside, of which molecules explore the entire volume. In that case, the confinement radius of molecules should scale with the focus size [40]. Conversely, when several condensates cluster without being liquid, molecules within the cluster should have the same confinement radius as in the situation of a single focus, if the clustered foci remain sufficiently distant. In an intermediate scenario, multiple solid foci might cluster and be sufficiently close for molecules to “jump” from a focus to another one: in that case, we expect to observe bimodal trajectories exhibiting confined diffusion in the first focus followed by 1 or several larger steps and confined diffusion in a second focus.

#####  **Diffusivity/Concentration Relation and Free Energy**


In the fast limit of binding/unbinding, there is a possibility that molecules exhibit confined motion within the condensate in the binding/PPPS model, in particular, if binding sites themselves diffuse. Indeed, theoretical work shows that the PPPS/binding model can be reduced to an effective description that is mathematically equivalent to the LPPS model, but with specific constraints linking its properties [58]. Thus, it can be difficult to discriminate the binding/PPPS models from an LLPS when monitoring only the exchange rate and internal mobility. Combining simulations and analytic approaches, Heltberg et al. derive two criteria that must be verified for the binding/PPPS models, allowing us to reject the binding/PPPS models when they are not satisfied. First, Heltberg et al. could establish a general relation between protein concentrations inside and outside foci versus their diffusivity for a wide range of parameters for the binding/PPPS models (the ratio of densities inside and outside the focus scales with (*D*_0_
*− D_b_*)/(*D_n_ − D_b_*), where *D*_0_ is the diffusion coefficient inside the condensate, *D_b_* is one of the binding sites, and *D*_n_ is the diffusion coefficient in the nucleus). As these observables are accessible by single molecule approaches, it is, thus, possible to assess whether the experimental point fits the prediction of the binding/PPPS model or not. Second, Heltberg et al. estimated the Gibbs free energy in the case of the binding/PPPS models by the relation: $U\left(r\right)=k_{B}T\ln\left[\frac{D\left(r\right)-D_{b}}{D_{n}-D_{b}}\right]$, where *D (r)* is the radial diffusion coefficient inside the condensate, *k_B_* is the Boltzmann constant, and T, the temperature. If the free energy does not follow this relation, the binding/PPPS models can be rejected.

#### 2.2.3. Additional Macroscopic Criteria

Other macroscopic criteria are also commonly used in literature to assess the physical nature of condensates but remain limited and some are prone to artefacts. Since they are used in some of the studies described in the following section, we briefly discuss them below.

#####  **Formation of Droplets In Vitro**


Several studies used purified proteins to quantify their ability to form liquid condensates versus solid aggregates in vitro. This approach was first used for heterochromatin protein 1 (HP1), a key element in heterochromatin formation [59,60,61], but also for repair proteins [62,63]. However, the concentration of proteins required for phase separation in vitro is often much larger than the one found in the nuclei [59]. Furthermore, the complex biochemical environment of the nucleus can either prevent or favor the formation of LLPS. Therefore, LLPS formation in vitro is neither necessary nor sufficient for their formation in vivo.

#####  **Dissolution upon the 1.6-Hexanediol**


The aliphatic alcohol 1.6-hexanediol was proposed as a tool to differentiate between liquid-like and solid-like assemblies in living cells [64]. Aliphatic alcohol 1,6-hexanediol disrupts weak hydrophobic interactions between IDRs that drive some LLPS [64]. Several in vivo studies in yeast and mammalian cells showed that 1.6-hexanediol treatment dissolves dynamic liquid-like assemblies, such as P-bodies or Rad52 foci, whereas solid-like assemblies, such as protein aggregates and cytoskeletal assemblies, are largely resistant to hexanediol [40,49,64]. When droplets are associated with chromatin, as is the case for LLPS in the context of DNA damage, 1,6-hexanediol treatment should be carefully interpreted. Indeed, it was shown that 1,6-hexanediol treatment rapidly slows down histones turnover, increases chromatin compaction [65] and impairs kinase activity [66].

#####  **Change in Light Diffraction Observed by Transmission**


Large condensate can be visible by bright-field microscopy, reflecting a change in refractive index that may arise from a distinct phase separated from the surrounding nucleoplasm. LLPS are expected to exhibit such a change in refractive index and are sometimes used for large foci [12].

Overall, here, we define three models (binding/PPPS/LLPS) and we give an overview of criteria to distinguish them. These models remain simplified. In reality, several parameters in each model can vary depending on the proteins studied and we can always imagine a situation for which a condensate is at the limit between two models.

## 3. Evidence and Limitations of Liquid Phase Separation at DNA Repair Sites 

In recent years, several studies investigated the physical nature of repair foci formed in response to DNA damage. In this section, we discuss the evidence and the limitations of liquid phase separation during the repair of DSB across different organisms. Figure 3 illustrates the choreography of repair proteins, focusing on the ones thought to form foci via LLPS in human and in *S.c*. yeast.

### 3.1. Technical Limitations

Assessing the physical nature of condensates can involve different experimental strategies, each having its limitations that are important to have in mind when interpreting experimental data. For instance, many studies used cell lines expressing fluorescently tagged proteins to follow their behavior and localization upon DNA damage. While some studies use endogenous tagged proteins, others are performed on transfected cells expressing tagged proteins in addition to their endogenous version under the control of ectopic promoters. Changing the physiological level of proteins expression might change condensate properties; it is, thus, essential to favor experiments using endogenous tagged proteins. It should also be borne in mind that the labelling of proteins may alter their biochemical properties and that the functionality of tagged proteins must be systematically evaluated. For example, the number and size of condensates formed by the repair factor TAF15 in living cells changes depending on the fluorophore used to tag it [67]. Finally, the paraformaldehyde fixation used for immunofluorescence or super resolution microscopy can strongly alter the appearance of LLPS, both enhancing or decreasing their numbers and sizes [40,67]. Conclusions relying only on data obtained from fixed cells should, thus, be interpreted with caution.

### 3.2. Evidence of Liquid Phase Separation at DNA Repair Sites

#### 3.2.1. PAR Chains, FUS, EWS and TAF15

Concerning LLPS in human cells, one of the earliest detectable events of the DNA damage response is the rapid increase in ADP-ribosylation signaling at the sites of damage [68]. PARylation is mainly triggered by the PARP family, the protein PARP1 having the most important PARylation activity (90% of the PARylation). In vitro, it was observed by TEM (transmission electronic microscopy) that isolated PAR chains accelerate the aggregation of low complexity domains (LCD) [12]. Indeed, when incubating PAR chains with a model peptide or with proteins of the FET family (FUS, EWS and TAF15) containing LCD, they found that spontaneous aggregates formed in vitro and these aggregates were consistently larger in the presence of PAR. Due to two phosphate groups per ADP-ribose unit, PAR is a highly negatively charged polymer, which during the peak of PAR production recruits a large amount of PAR-binding proteins by multivalent electrostatic interactions, including FUS, EWS and TAF15 [68]. The formation of aggregates in vitro indicates that PAR chains catalyze the collapse of proteins containing LCD, but it does not allow for distinguishing between a polymer and a liquid phase. Combining these observations in living cells, they found that PAR-seeded liquid demixing resulted in the rapid, yet transient and fully reversible assembly of FUS, EWS and TAF15 proteins at DNA breaks. In the absence of dePARylation, the local increase in light diffraction, one of the hallmarks of LLPS, is more pronounced at damaged sites. Overall, they conclude that PARP1 does not form LLPS by itself, but PAR chains initiate the formation of LLPS for several intrinsically disordered proteins (FUS, EWS and TAF15) at DNA break sites, a process tightly regulated upon dePARylation by PARG (see Figure 3).

#### 3.2.2. 53BP1 and p53

FUS foci are dynamic and, after PAR removal by PARG, rapidly dissolve, which likely serves to allow access by repair factors to damaged DNA [63]. It was shown that FUS foci can be replaced by foci formed by the repair protein 53BP1, which forms liquid-like condensates via interactions with long non-coding RNA transcribed near the double-strand breaks [51,52], committing the break to repair by non-homologous end joining. Using live cell microscopy and CRISPR/Cas9-mediated endogenous protein tagging, Kilic et al. showed that 53BP1 foci are dynamic, show droplet-like behavior and undergo frequent fusion and fission events. Based on live cell imaging, it was proposed that 53BP1 acts as a scaffold for further repair proteins such as the tumor suppressor p53 [51]. 53BP1 interacts with chromatin to enhance c-NHEJ and suppress HR.

Of note, there are no data in living cells at the single molecule level to investigate the internal dynamics of foci formed by 53BP1 or proteins from the FET family; similarly, the diffusion behavior at the focus boundary or the effect of over-expression have not been investigated yet. Thus, further work will be determinant to assess the liquid natures of these condensates at the microscopic level. In addition, only the recent study by Kilic et al. used a cell line with a CRISPR/Cas9-mediated endogenous 53BP1 protein tagging, while other studies were performed on transfected cells.

#### 3.2.3. Rad52 Proteins in Yeast

Two studies using different microscopy approaches showed that in *Sc.* yeast, Rad52 foci exhibit several LLPS properties. Using conventional microscopy, Oshidari et al. showed that Rad52 foci have a spherical shape and flatten against the nuclear envelop or damage-inducible microtubule filaments (DIMs) [49]. In addition, in the presence of multiple DSBs, multiple Rad52 fuse into a repair center droplet via the action of petite DIMs. Using SPT of Rad52 molecules in living yeast cells, Miné-Hattab et al. showed that Rad52 foci are highly dynamic with a constant rearrangement of molecules within the focus. Of note, based on live PALM (photo activable localization microscopy), they estimated the size of Rad52 foci as 120 nm, thus, the sphericity measured in Oshidari et al. is due to the diffraction light and cannot be used as a meaningful criterion for LLPS. Comparing trajectories of Rad52 within foci with their size, Miné-Hattab et al. found that Rad52 molecules explore the entire volume of the focus. Thanks to the power of Sc. yeast genetics, it is possible to insert a single or several controlled DSBs in the yeast genome. Miné-Hattab et al., thus, compared the size of Rad52 foci induced by a single DSB versus two DSBs.: They found that Rad52 foci resulting from two DSBs are twice as large in volume than the ones induced by a unique DSB and the Rad52 confinement radius scales accordingly. Furthermore, Rad52 particles spend much more time inside the focus than would be predicted from their diffusion coefficient based on the binding model (Heltberg et al., 2021).

SPT experiments also suggest that SUMOylation might modulate the liquid properties of Rad52 foci. Indeed, when all Rad52 molecules are SUMOylated, their motion inside foci remains confined but becomes significantly slower than wild type Rad52. In addition, foci of SUMOylated Rad52 are denser [40]. Miné-Hattab et al. proposed that foci formed with SUMOylated Rad52 proteins transition to a gel-like condensate where most of the molecules are still free to explore the whole focus but much slower than in wild type cells. 

Altogether, these two studies provide strong evidence that Rad52 foci have LLPS properties. However, using conventional microscopy to measure the intensity of the Rad52 nuclear background, Miné-Hattab et al. found that Rad52 background concentration increased, as predicted for a client of a droplet. It is not known which protein in yeast would drive the initial formation of LLPS at DSB sites.

#### 3.2.4. SSB in *E*. *coli*

A recent study proposed that SSB protein molecules form liquid–liquid phase-separated condensates [62]. Their results are based on in vitro experiments. Efficient phase separation requires all structural modules of SSB and is regulated by the specific interaction between the CTP (C-terminal peptide), as well as the stoichiometry of available SSB and ssDNA. These results are not consistent with observation in *Sc.* yeast with its functional analogue Rfa1 showing that Rfa1 foci are not liquid. It may be because SSB contains an intrinsically disordered linker (IDL) that is not well conserved across evolution and might play an important role in the liquid properties of SSB foci. More evidence in vivo will be necessary to clarify these results.

### 3.3. Limitation of the LLPS Model in the Context of DSB Repair 

There is still a long list of repair proteins forming foci for which the physical nature remains unknown. The liquid phase separation model has received so much attention that among the studies on repair condensates, the large majority present evidence of LLPS. It seems that not being an LLPS would be considered a negative result although foci formation could also be explained by alternative models. It also gives a false impression as if repair condensates were all formed through liquid phase separation. In the following section, we highlight several studies presenting alternative mechanisms for the formation of repair foci.

#### 3.3.1. RPA Foci Do Not Exhibit LLPS Properties in Yeast

In *S. cerevisiae* yeast, it was shown that the ssDNA-binding protein Rfa1, a subunit of the RPA complex, does not form an LLPS, unlike its partner Rad52. Comparing the dynamics of Rad52 and Rfa1 by SPT, Miné-Hattab et al. observed a very different diffusive behavior for both proteins in response to a single DSB. While Rad52 is highly dynamic with a diffusion coefficient ~6 times higher than damaged chromatin, Rfa1 follows anomalous motion with the same diffusive properties as damaged chromatin [40]. Moreover, Rfa1 foci are not sensitive to 1.6-hexanediol treatment while Rad52 foci are partially dissolved. Overall, although Rfa1 and Rad52 foci have similar macroscopic appearance, the formation of Rfa1 condensates is not consistent with an LLPS and can be explained by a simple binding model on the ssDNA tail of the DSB.

#### 3.3.2. Rad51 Forms Foci and Filaments at DSB Sites, Inconsistent with LLPS 

The ScRad51, as well as hRAD51 and its functional homolog RecA in bacteria, exhibits unique structures that are inconsistent with LLPS. Through biophysical studies and EM images, it is shown that Rad51 would remove RPA and form a right-handed helix structure along the ssDNA whose length can be up to several micrometers (Liu et al. 2011, submitted). Consistently, RecA fused to fluorescent proteins was shown to form long filaments or bundles in the presence of the endogenous untagged protein in response to DSB induction in living bacteria [69,70]. Using an endogenous fully functional GFP Rad51 fusion, Liu et al. observed a Rad51-DNA complex in living yeast forming foci as well as dynamic filaments (submitted manuscript). More than 30% nuclei form foci after inducing a DSB for 2 h, while around 50% nuclei present a Rad51 filaments after 4 h I-*Sce*I DSB induction, suggesting the Rad51 foci observed after 2 h are the initiation of Rad51 filaments. Emerging filaments are extremely dynamics and can adopt a variety of shapes. Time-lapse microscopy revealed that long Rad51 filaments can undergo several cycles of compaction and re-extension allowing a rapid and robust exploration of the nuclear volume. When compacted, Rad51 filaments could be either collapsed Rad51-DNA structures in which Rad51 depolymerized or entangled Rad51 filaments. Therefore, though sometimes forming foci, the complexes formed on damaged DNA are inconsistent with LLPS, and further models are required to explain its dynamics.

#### 3.3.3. Internal Architecture of Repair Foci

Although most of the repair factors forming foci are described as a homogenous sphere under conventional microscopy, super resolution microscopy has recently revealed the existence of more complex foci architecture [71,72,73]. In particular, following the induction of multiple DNA damages by micro-irradiations, a repair “super-focus” (occasionally more than 2 μm in diameter) is formed. They contain the recruited 53BP1 in the outer shell, HP1 mostly in the center [74], phosphorylated moieties of histone H2AX (γH2A.X) all throughout its volume. These super-foci, also referred to as “micro-domains” are formed of four to seven 53BP1 sub-foci of about 100 nm in diameter, similar to a pearl necklace. These sub-foci are maintained together by RIF1, which is located between each 53BP1 sub-foci. 53BP1-sub foci failed to mature to circular MDs in RIF1 depleted cells but not in SHEiding depleted cells, indicating that RIF1 has a unique role in stabilizing chromatin topology initiated by the formation of 53BP1-sub-foci [71]. Ten minutes after irradiation, the c-NHEJ factor XRCC4 is observed in the center of the 53BP1 structure, while 1h post-irradiation, BRAC1 or Rad51 can be found there. The internal architecture of foci might also depend on the numbers and the nature of DNA damage. It is possible that repair foci formed in response to an individual double-strand break, or a small number of DSBs exhibit no discernible internal organization, while a repair focus formed in response to a cluster of DSBs forms a more complex architecture with a 53BP1-rich outer layer. Emerging techniques allowing the induction and visualization of a single DSB or SSB (single strand break) will help clarifying these open questions [75].

Overall, it is clear that several kinds of condensates co-exist at damaged sites with different physical natures: further work will be necessary to dress a comprehensive picture of how the different players organize each other. Table 1 summarizes the different repair proteins for which the physical nature of foci formed upon DSB was studied.

## 4. The Origin of Phase Separation and Their Possible Functions

### 4.1. Origin of Phase Separation

The formation of LLPS is driven by weak multivalent interactions between macromolecules. Multivalent interactions are molecular associations between multiple binding sites on the interacting molecules. Within an LLPS, weak multivalent interactions usually involve folded protein domains, intrinsically disordered regions (IDR) [76], low complexity domains (LCD) [77], nucleic acids [52] or chromatin. IDR are protein regions that lack a fixed and well-defined 3-dimensionnal structure. LCDs in protein sequences are unusual regions made up of only a few different types of amino acids. LCDs were originally thought to be only flexible linkers used to separate the structured and functional domains of complex proteins. However, they can also form secondary structures, such as helices and even sheets with specific functions [78]. In recent years, with the increasing interest concerning LLPS in cell biology, the role of IDR and LCD appears to be much more important than initially thought.

#### 4.1.1. Liquid Phase Separation by RNA during DNA Repair

RNA innately mimic LCD and are found in different kinds of LLPS, such as P-bodies [79], stress granules [80] or repair foci in human cells [52]. In human cells, it was proposed that DSB-induced transcriptional promoters drive RNA synthesis and stimulate phase separation of repair proteins [52]. More specifically, damage-induced long non-coding RNAs (dilncRNA) synthesized at DSBs by RNA polymerase II (RNAPII) appear to be necessary for the formation of repair foci such as 53BP1 in human cells.

#### 4.1.2. Liquid Phase Separation by Poly(ADP-Ribose)

PAR is a polymeric molecule that shares several features with RNA, including a high negative charge and structural diversity (size, chain length and branching complexity) [63]. PARP1 proteins are not forming an LLPS per se but it is rather proposed that they initiate the formation of LLPS by other proteins binding the PAR chains. As complexes form between PAR molecules and proteins, the entropic cost of confining the PAR-protein complexes into the condensed phase is lower than the cost of confining individual components. This difference in entropic cost due to multivalency partly drives phase separation. Phase separation occurs when the affinity between PAR and protein is higher than the affinity between cellular fluid with either macromolecules [81].

#### 4.1.3. Liquid Phase Separation by Repair Proteins

In *Sc.* yeast, there is no PARylation upon DNA damage and no or very little damage-induced RNA formation. However, some repair proteins exhibit hallmarks of LLPS, such as Rad52 [40,49]. Rad52 behavior is more consistent with the one of an LLPS “client” [40], and the mechanisms driving the formation of Rad52 LLPS remain unknown. In human cells, repair proteins of the NHEJ or HR pathway are more often described as clients of LLPS already formed during earlier stages of the DDR. However, more studies will be necessary to reveal the physical nature of each repair factor condensate and how different kinds of condensates co-exist.

### 4.2. Possible Functions of Liquid Phase Separation

The formation of condensates via LLPS presents some advantages, which are presented in the following section and summarized in Figure 4.

#### 4.2.1. Efficiently Create Micro-Environments with High Local Concentration of Specific Proteins

Cells are highly spatially organized and contain millions of proteins. Such large-scale organization means that proteins are often produced far from the site of their function. A simple Brownian diffusion of proteins might not be sufficient to bring them to their sites of function. Cells have, thus, evolved mechanisms to form sub-compartments, maintain and dissolve them within the proper time-window. In the nucleus, compartments are all membrane-less. Inside, specific proteins are more highly concentrated and such enhanced concentration is hypothesized to help the proteins coordinate and collectively perform their function. While the different models of condensates allow the formation of highly concentrated foci, the LLPS model has unique properties that might be crucial for their function. For example, LLPS creates a micro-environment inside which some proteins are excluded while others are not, unlike the binding and the PPPS models [41].

Moreover, the kinetics of LLPS formation are tightly regulated by the diffusion coefficient of its components, unlike the PPPS/binding model [58]. Indeed, Heltberg et al. estimated the time it takes for a molecule to find the focus from the edge of the nucleus and to find the target from the focus boundary. Using both analytic and simulation approaches, they found that in the LLPS model, this search time exhibited a clear minimum corresponding to the experimental focus size; in contrast, the search time in the PBM did not have a clear local minimum. In other words, foci formation via LLPS may act as a funnel accelerating the search by repair molecules for the DSB. In the PPPS/binding model, such an improvement is negligible unless binding sites themselves have a fast diffusive motion. In the case of DNA damage, repair foci are formed for short periods of time to repair DSB and exhibit a very coordinated choreography involving many repair factors [6,36]. In this case, the speeds of both focus formation and target finding are important for rapid repair, but long-term stability of foci is not needed.

#### 4.2.2. Buffer the Concentration of Proteins

In the LLPS model, increasing the concentration of its components results in a larger volume fraction of the dense phase (with constant dense-phase concentration) and a smaller volume fraction of the dilute phase, i.e., the concentration in the dilute phase remains constant. Alternative models, such as the binding or the PPPS/binding, do not produce this buffering behavior. Thus, it was proposed that when protein levels increase, LLPS might be a mechanism to store the excess of proteins while keeping the dilute phase at a constant concentration.

#### 4.2.3. Reshape Chromatin at Damage Sites 

Evidence is emerging that LLPS can exert force, locally reshaping cellular architecture. LLPS were proposed to be beneficial during the origins of life, when molecules would have been otherwise diluted and unfolded [82]. Furthermore, the fusion of chromatin-targeted LLPS can bring distant genomic sites into proximity [82]. Bridging distant genomic sites is per se a property of the PPPS model; this is, thus, not an advantage specific to the LLPS. However, the creation of low chromatin density regions is specific to the LLPS and might play a role in allowing repair proteins to access to the damaged site.

#### 4.2.4. LLPS in the Origin of Life

The LLPS provides unique micro-environments that concentrate specific molecules without any membrane. LLPS were proposed to be beneficial during the origins, when molecules would have been otherwise diluted and unfolded [83]. Modern cells are thought to arise from a more primitive form of compartmentalization on early Earth (i.e., a protocell), which provided a primitive system the ability to concentrate and segregate molecules in the absence of compartment with membranes. In vitro, LLPS could be formed through primarily energetically favorable processes (such as entropic effects and hydrogen bonds [84]) rather than through the production of energy intensive covalent bonds. These LLPS can be composed of simple heterogeneous polymer systems resembling synthetic products from early Earth, suggesting that LLPS compartments may have been able to form easily at that time.

## 5. Conclusions

During the last 10 years, several studies focused on the physical nature of repair condensates. While most of the studies revealed suggested that repair foci behave sim-ilar tolike LLPS, the diffusion and structure of some repair proteins is not consistent with LLPS. It is possible that several kinds of condensates with different physical na-tures co-exist at DSB, simultaneously or one after another. It will be exciting in the fu-ture to study their choreography combining advanced imaging and genetics.

Our increasing understanding of condensate function provides opportunities to target these structures in the treatment of disease. Defects in condensates were implicated in numerous diseases. Most of these are associated with neurodegenerative diseases but defective condensates were also implicated in various cancers [85,86]. In the future, modulation of condensates might be a novel approach to treat pathologies with defective condensates.

## Figures and Tables

**Figure 1 genes-13-01846-f001:**
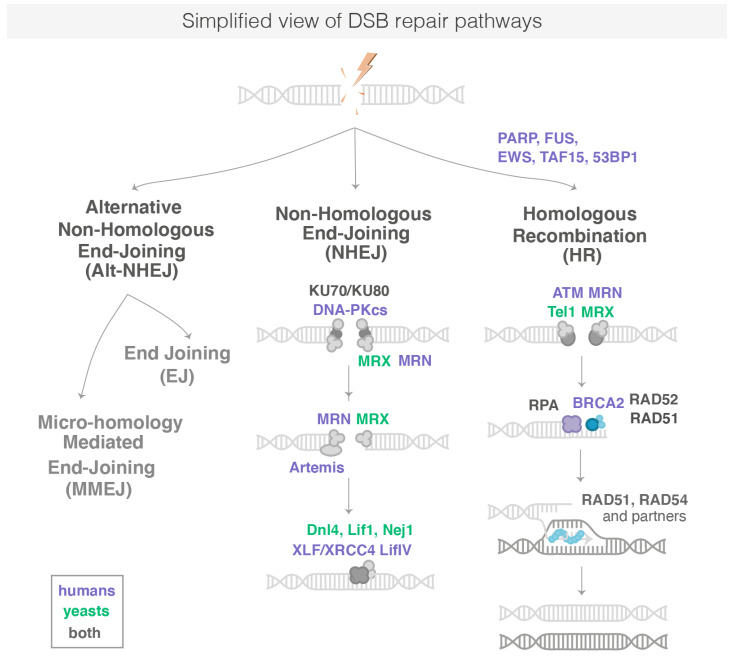
Simplified representation of DSB repair pathways. DSB are repaired by HR, c-NHJE or Alt-NHEJ. Proteins in bold are known to form repair foci in cells. The sub-pathways of the Alt-NHEJ are not represented here. NB: Proteins common between Sc. Yeast and human are written in grey with the nomenclature used for human proteins.

**Figure 2 genes-13-01846-f002:**
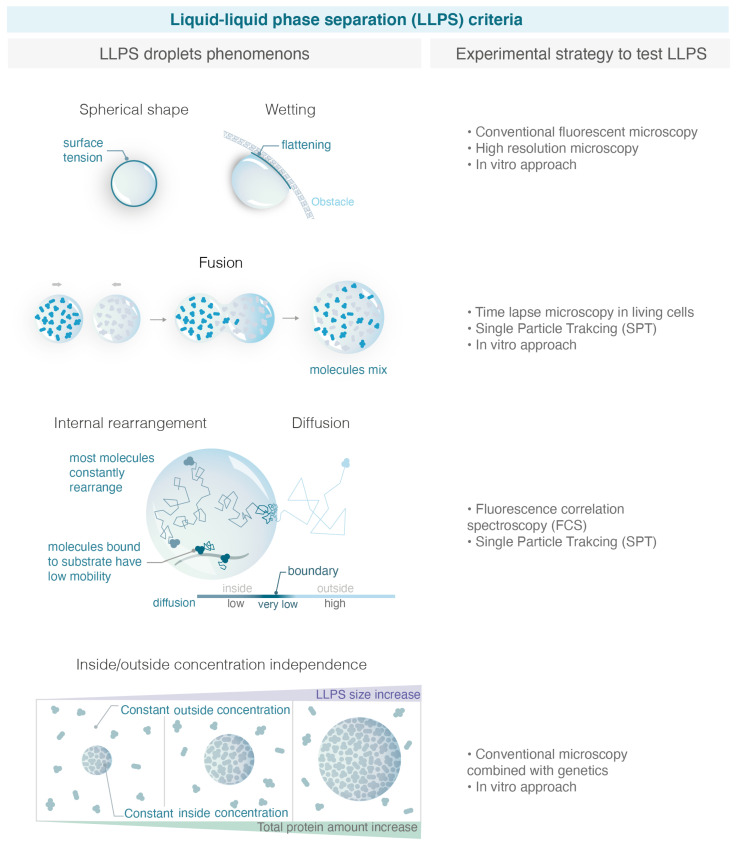
Criteria used to define LLPS and experimental strategies to test them. With the increase in studies claiming that condensates are liquid phase separation, it is important to define properly robust macroscopic and microscopy experimental criteria able to distinguish LLPS from alternative models. On the left, the figure illustrates the criteria explained in the text, and on the right, the corresponding experimental strategies allowed to test them. (Illustration by Miné-Hattab and Olga Markova).

**Figure 3 genes-13-01846-f003:**
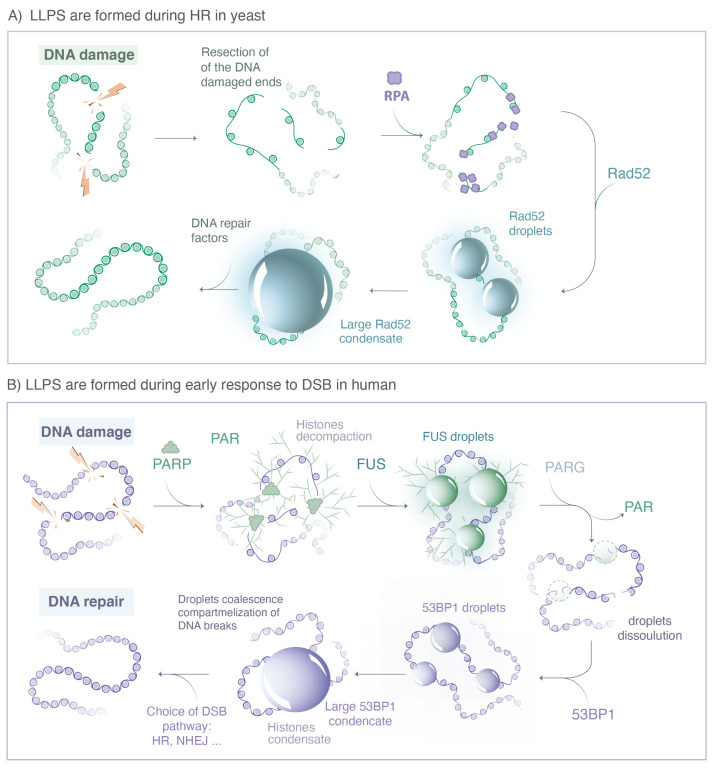
Repair proteins known to form LLPS during DSB repair. (**A**) In yeast, when a DSB is repaired by HR, the 5′ ends of the DNA break are resected by nucleases to yield 3′ single-stranded DNA (ssDNA) tails that are rapidly coated by the RPA complex. The Rad52 protein, the functional analog of human BRCA2 in yeast, then stimulates the removal of RPA. The ssDNA-binding protein Rfa1, a subunit of the RPA complex, and Rad52 form both visible foci at damaged sites; however, they exhibit very different behavior at the microscopic level. While most Rfa1 molecules are bound to the ssDNA, Rad52 molecules exhibit several hallmarks of LLPS. Changing the concentration of Rad52 behaviors is consistent with a client of an LLPS. (**B**) In human cells, immediately after DNA damage, the PARP proteins family is recruited at the sides of single and double strand breaks, inducing PARylation of histones and auto-PARylation. ATM and MRN complexes are rapidly recruited at damaged sites, likely through a binding mechanisms without forming LLPS. The FET family proteins (FUS, EWS, TAF15) binds PAR chains formed at damaged sites. PAR chains initiate the formation of FUS, EWS and TAF15 condensates, which show several hallmarks of LLPS. After dePARylation by PARG, these droplets are dissolved and replaced by 53BP1 condensates, also exhibiting hallmarks of LLPS. (Illustration by Miné-Hattab and Olga Markova).

**Figure 4 genes-13-01846-f004:**
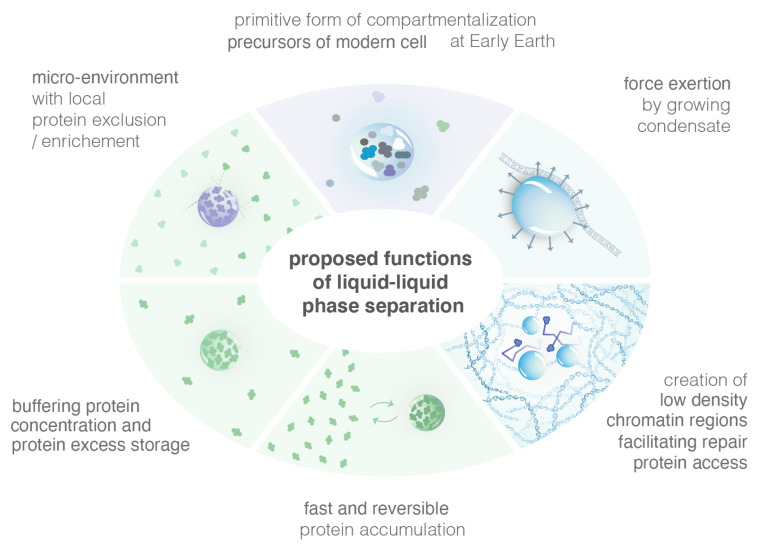
Illustration of several proposed functions of LLPS.

**Table 1 genes-13-01846-t001:** List of repair proteins for which the physical nature of foci formed upon DSB was studied. For each protein, we precisely list which criteria were used to confirm or reject the LLPS model. The sign “-“ indicates that the criteria was not tested.

Proteins	Organisms	Physical Nature of the Condensate	Sphericity	Fusion	Internal Dynamics	Diffusion across Boundary	Concentration Dependence
SSB	*E. coli*	LLPS	-	-	Yes (FRAP in vitro) [62]	-	Yes in vitro [62]
RPA1	*S.c* yeast	Not LLPS	-	-	No (Identical to chromatin) [40]	No [40]	-
Rad52	*S.c* yeast	LLPS	Yes [49]	Yes [40,49]	Yes [40]	Yes [40]	No [40]
PARP-1	Human	Initiate only LLPS	-	-	-	-	-
FUS	Human	LLPS	Yes [12,67]	Yes [12]	-	-	-
EWS	Human	LLPS	Yes [12,67]	Yes [12]	-	-	-
TAF15	Human	LLPS	Yes [12,67]	Yes [12]	-	-	-
53BP1	Human	LLPS	Yes [12]	Yes [12]	-	-	-
p53	Human	Client of LLPS [51]	-	-	-	-	-

So far, studies showing complex foci architecture with an 53BP1 outer layer do not discuss their physical nature; conversely, studies claiming the LLPS properties of 53BP1 foci do not have the resolution to see their internal structure. It will be important in the future to investigate the link between the internal organization of foci, their physical nature and their functions.

## Data Availability

The study did not report any data.

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
