# Peer review of "Repair Foci as Liquid Phase Separation: Evidence and Limitations"

_genes, 2022, doi:10.3390/genes13101846_

Round 1

Reviewer 1 Report

The manuscript “Repair foci as liquid phase separation: evidence and limitations” by Miné-Hattab et al. is devoted to the actively developing fundamental problem of the existence and functioning of reparation "foci", which are formed in response to DNA damage. 

1.       Lines 111, 138, 151, 363, 564 incorrect reference format

2.       Line 380 please remove the extra indefinite article: “a situation for which a condensate is a at the limit between 2 models”

3.       Lines 418-422 font does not match

4.       Lines 435-436 the sentence “The formation of aggregates in vitro indicates that PAR chains catalyze the collapse of proteins containing LCD...” is not clear. How can the PAR polymer catalyze something, and what kind of protein collapse are the authors talking about?

5.       Line 499: “Repair proteins that do not form LPS in response to DSB” probably, LLPS?

6.       Line 552: “(IDR) are proteins region” I guess, it should be “IDR are protein regions

7.       Lines 554 and 558 probably, LCR should be replaced with LCD?

8.       Line 571 “PARP1 proteins is not forming” should be “PARP1 proteins are not forming”

A general note to the whole text: the review is embarrassingly poorly illustrated. It may be worth adding the scheme of homologous recombination in the context of condensate formation to section 1.1 and the scheme of functions of liquid phase separation to section 4.2.

Figure 1 can be divided into sections A, B, C, etc., to make it easier to refer to in the text. The authors could also add minor criteria 2.2.5 – 2.2.12 to the figure 1 for clarity and easier perception.

Reviewer 2 Report

In the manuscript by Mine-Hattab et al., the authors describe the current state of research on whether DNA damage-dependent foci might be formed by liquid phase separations or not. The text first provides background on DNA damage response foci or “condensates” then goes on to describe the possible biophysical nature of types of foci including defining liquid-liquid phase separation (LLPS) and methods to distinguish between the biophysical types, and lastly describes the evidence suggesting LLPS as a mechanism (or not) for formation of specific foci. The authors are clearly experts in this expanding area of research and the subject of the paper is very interesting. They also do a reasonably good job of conveying complex biophysical principles in a way that most scientists can grasp. However, organization and clarity of the background section on formation and relevance of DNA damage response-dependent foci could be considerably improved, which would contribute to understanding and clarity of the material to follow. In addition, there are several less important issues and the writing (grammar and spelling) that need to be addressed.

Major Points

1. Since the emphasis of the review is on the biophysical nature of repair foci, the authors should do a more thorough job in summarizing and clarifying the research on the stepwise recruitment of factors to double-strand breaks (and potentially other repair foci where relevant). In this regard, the entirety of Section 1 is not very clear and could be better organized, perhaps adding some figures to help with clarity. The authors move back and forth between the yeast and mammalian models, which also adds to the confusion. Perhaps it would be preferable to clarify these models separately in the text. Another concern is that the authors fail to mention the alternative non-homologous end-joining pathway and whether factors involved in this pathway are involved in foci that form at double-strand breaks. Among the other things that need to be clarified; 1) whether there are different types of foci and the critical composition of each, 2) number of breaks and protein molecules in a typical focus, 2) order of recruitment of factors at least for the foci discussed further in the manuscript.

2. In section 1, the authors rightfully focus early on PAR activity by PARP1 in relation to recruitment of factors to foci, but do not clearly represent the multi-faceted role of PARP1 in various damage response and repair pathways. In particular, they appear to gloss over the role of PARP1 in repair of single-strand breaks compared to double-strand breaks and how this affects experiments involving damaging treatments that create both. Thus, this adds to the confusion mentioned above that undermines the remainder of the paper.

3. It seems that one of the best methods to distinguish LLPS from other condensate types is the use of single molecule microscopy to perform single particle tracking (SPT). In my opinion, the authors should expand this section somewhat by describing in more depth these measurements.

4. (section 3.2.1) While the authors previously mention FUS as a possible factor in DDR foci, the possible roles for EWS and TAF15 in the DDR are not clear and really not detailed here. If there is more evidence for their actual participation in DDR, it would be helpful to mention it here and/or in the background (section 1).

Specific/Minor Comments

1. (figure 2) The bottom panel of this figure seems to be slightly misleading. In my understanding, the choice of double-strand break repair pathway occurs before formation of 53BP1 foci. Once 53BP1 foci are apparent, HR is inhibited and non-homologous end joining is typically employed (as is mentioned in the text). If this is the case, then the figure should be adjusted accordingly. Also, in the top panel, the word “condensate” is misspelled.

2. Some typographical and grammatical errors occur throughout the manuscript. Although the majority of these are minor and do not affect the concepts communicated in this review, the manuscript should be carefully proofread to correct these errors. For example (lines 139 and line 140, it should be “dynamics are” instead of “dynamics is.” Also, the sentence “Indeed, it has been shown that 1,6-hexanediol rapidly slows down the dynamics of histones and condensates chromatin both in living cells and in vitro” is grammatically incorrect and/or confusing.

3. (lines 34-35) “Germline” mutations in DNA repair genes lead to cancer predisposition syndromes, while somatic mutations in repair genes may or may not contribute to cancer development.

4. (lines 40-42) The stated division of labor between NHEJ and HR in vertebrates is pretty misleading. Most research says the repair choice is mainly based on the cell cycle phase. So replicating cells with different cell division times would show different percentages based on the time spent in G1 versus S and G2 phases. It is much better to clearly convey the relative cell cycle phase specificity of the pathways instead of providing these specific percentages which would not apply to all cell models.

5. (lines 79-80) Referring to the processing done during NHEJ (even canonical NHEJ) as “minimal” is misleading. NHEJ processing involves trimming of “dirty” ends, removal or addition of nucleotides by a variety of factors and ligation. Maybe “resection” of double-strand breaks during canonical NHEJ is limited, although it can be more extensive prior to alternative NHEJ.

6. (lines 130-131) The authors suggest that, in mammalian cells, DNA-PK/Ku70 molecules are barely detectable at DSB sites. This contradicts some findings and the fact that Ku70 forms a tight complex with Ku80, a complex that is rapidly recruited to some if not all double-strand breaks. This needs to be clarified further.

7. (line 419 and 470) Define the acronyms STORM, PALM and SIM.

8. (lines 229-231) The protein markers used to follow droplet fusion in these experiments should be defined, at least to lead the readers to the idea that these factors or particular types of foci will likely be forming via LLPS.

9. (line 490) The term “SSB” should be replaced by “SSB protein molecules” to differentiate the meaning from an acronym for single-stranded breaks.

10. (lines 111, 151-152, 363-364) In several instances, citations have not been formatted correctly.
